# Herbal-Based Cosmeceuticals and Economic Sustainability among Women in South African Rural Communities

**Peter Tshepiso Ndhlovu** [1] **, Abiodun Olusola Omotayo** [2] **, Adeyemi Oladapo Aremu** [1,2,*] **and Wilfred Otang-Mbeng** [3]

[1]   Indigenous Knowledge Systems (IKS) Centre, Faculty of Natural and Agricultural Sciences, North-West University, Private Bag X2046, Mmabatho 2745, North West Province, South Africa; Tshepiso.Ndhlovu@nwu.ac.za

[2]   Food Security and Safety Niche Area, Faculty of Natural and Agricultural Sciences, North-West University, Private Bag X2046, Mmabatho 2745, North West Province, South Africa; omotayoabiodun777@gmail.com

[3]   School of Biology and Environmental Sciences, Faculty of Agriculture and Natural Sciences, University of Mpumalanga, Private Bag X11283, Mbombela 1200, Mpumalanga Province, South Africa; Wilfred.Mbeng@ump.ac.za

\*   Correspondence: Oladapo.Aremu@nwu.ac.za; Tel.: +27-18-389-2573

**Abstract:** Access to natural resources in the immediate environment is an essential factor that contributes to livelihood in many rural areas. In the current study, we explored the economic potential(s) of the natural herbal-based cosmetic and cosmeceutical enterprise for the welfare of the Vhavenda women. A purposive sampling technique was used to collect data from 79 Vhavenda women and analysed with descriptive and inferential statistics (Tobit regression) as well as budgeting analysis. The majority (61%) of the participants were married with an average household size of five members. Additionally, 39% of the participants were already ageing with an average age-group of 56–70 years. The majority (44%) of the participants were not formally employed while the monthly average total income of R1841.01 (107.37 USD) was recorded with an average per capital expenditure of R1438.42 (83.89 USD). A budgeting cost ratio of 1.28 was recorded, which indicates that for every R1.00 (0.057 USD) invested in the herbal-based cosmetic and cosmeceutical production, an expected return of R1.28 (0.073 USD) was forecasted. Tobit regression results indicated that the determinants of the income of participants were experience level ($p < 0.01$), religion affiliation ($p < 0.05$) and consumption expenditure ($p < 0.01$) among others. Thus, a conscious, introspective and intentional look into this marginalised herbal-based cosmetic and cosmeceutical enterprise as a panacea for improved income and welfare of rural South Africans should be considered.

**Keywords:** budgeting analysis; rural household income; medicinal plants; women; sustainability; Tobit regression

---

## 1. Introduction

Plants are indispensable to the existence of humans, as they play several significant roles in maintaining the quality of life, nutrition and medicinal needs (Noorhosseini et al. 2019; Beltreschi et al. 2018; Lawal et al. 2020). In addition, natural-based cosmetics contribute to an increased income for rural households and communities as well as their overall welfare. Shackleton et al. (2007) argued that access to and the use of natural resources is important for poverty alleviation. Antignac et al. (2011) indicated that currently, there is an increasing demand for cosmeceutical products that contain natural and/or organic ingredients, especially in North

America and Western Europe. Due to the increasing safety concerns associated with the use of synthetic cosmetics, there is a renewed interest in exploring natural resources, especially plants, for herbal-based cosmetics and cosmeceuticals (Antignac et al. 2011; Mahomoodally and Ramjuttun 2016; Lall and Kishore 2014). Therefore, accessing and trading herbs for natural-based cosmetics has the potential to enhance the local economy. Furthermore, herbal-based cosmetics and cosmeceuticals are an important socio-cultural heritage in many rural communities. For instance, in Pakistan, herbs are used for preserving and enhancing the beauty and personality of human beings (Ahmad et al. 2008).

South Africa is known to have a remarkable plant diversity which is largely untapped in terms of its potential for medicinal and cosmetic purposes (Van Wyk 2011). Regardless of being a significant part of the natural and social assorted variety, only a few plants are currently marketed on commercial scale. According to Mander et al. (2007), the trade of medicinal plants in South Africa is a large and expanding industry, with estimated R2.9 billion contribution to the economy. However, the pricing structures fluctuate across markets and over-time as the cost of harvesting plant species varies depending on a gatherer's access to the resources and the proximity of markets to the harvesting sites (Williams et al. 2007). In South Africa, many plants are used by local communities as sources of cosmetics, medicines and food (Van Wyk 2015). Likewise, among the Vhavenda in Vhembe District Municipality of Limpopo province, South Africa, plants remain popular and well-enriched in their tradition and culture for different purposes (Rampedi and Olivier 2013). These diverse indigenous plants offer the rural communities valuable environmental resources that could be used to maintain the welfare of people living in this area.

Generally, the use and the impact of environmental resource has some degree of influence on household welfare among the indigenous communities in Southern Africa (Ntuli and Muchapondwa 2017; Thondhlana and Muchapondwa 2014). For instance, environmental income from resources such as cosmetics, fire wood, medicinal plants and wild fruits contribute about 20% to the total income of the indigenous San and Mier rural communities of Kalahari drylands in South Africa (Thondhlana and Muchapondwa 2014). Even though the actual financial benefits from the sale of plants and their products in the informal sector remain largely undocumented and form part of the 'hidden' economy, it is generally known that medicinal plants and their products contribute substantially to the economy of local communities in South Africa (Botha et al. 2004; Makunga et al. 2008). Research efforts have often neglected the socio-economic context of herbal-based cosmetics and cosmeceuticals as well as its contribution to the local economy. Particularly, there is a dearth of information regarding the economic potential of herbal-based cosmetics and cosmeceuticals among the Vhavenda women in Vhembe District Municipality, South Africa.

Thus, this study evaluated the socio-economic characteristics, the welfare status (proxied by their income) of the Vhavenda women, the profitability level of the herbal-based cosmetics and cosmeceuticals and the factors influencing the income generated from herbal-based cosmetics and cosmeceuticals in the study area. We intent to advocate and motivate timely policy intervention using the appropriate channels in order to drive and sustain funding and support for research on herbal-based cosmetics and cosmeceuticals. The current study is geared towards addressing the following research questions:

1. What are the socio-economic characteristics of the women involved in the trading of herbal-based cosmetics and cosmeceuticals in the study area?
2. What are the factors influencing the income generated from herbal-based cosmetics and cosmeceuticals in the study area?
3. Is herbal-based cosmetics and cosmeceuticals a profitable venture and to what extent is there profitability among women in Vhembe district area?

## 2. A Brief Literature Review on Herbal-Based Cosmetics and Cosmeceuticals

The history of herbal cosmetics in European and Western countries consists of very dark phase in the late 6th century when different concoctions and pastes were used to whiten the face; this practice remained popular for over four hundred years (Chaudhri and Jain 2014). The early mixtures that were used in Europe for this purpose were so potent that they sometimes led to paralysis, strokes or death (Mansor et al. 2010). Globally, herbal-based cosmetics are gaining popularity as evidenced by rapidly increasing global and national markets of herbal products (De Janvry and Sadoulet 2001). The global therapeutic market was worth USD 550 billion and USD 900 billion in 2004 and 2009, respectively (Butler et al. 2014). The present demand for herbal-based cosmetics and cosmeceuticals is USD 14 billion per year and is projected to increase to USD 5 trillion by 2050 (Jeelani et al. 2017). Turnover rate for the herbal and Ayurveda industry in Sri Lanka, which is regarded as one of the leading markets, is approximately USD 2.5 billion per year (Booker 2014). The production and marketing of herbal products have been growing fast in many major markets including Germany, USA, France, China, Italy, Japan, UK and Spain (Mafimisebi et al. 2013).

According to Bilal et al. (Bilal et al. 2016), herbal cosmetics are highly valued and nearly three-quarters of the cosmetics and cosmeceuticals that are used globally are discovered from local plants. In addition, about 25% of modern cosmetics are derived from plants (Amit et al. 2010). Many synthetic cosmetics and cosmeceuticals are based on prototype mixtures which were isolated from plants. Thus, herbs are a potential source of therapeutics and have attained a significant role in health systems globally. However, communities in developing countries lack sufficient information on the social and economic benefits that could be derived from the industrial utilization of traditional plants. As indicated by Alves and Rosa (2007), apart from the use of these plants for local healthcare needs, information has to be available about market potential and trading possibilities.

Socio-economic factors such as household income, household size, the age, employment, and marital status and the educational attainment of the household head significantly affect the socio-economic status of households (Delpeuch et al. 1999; Ndayambaje et al. 2012; Nguyen and Nguyen 2019). The welfare function may differ across the rural households and across circumstances, indicating that the same amount of real income may produce different levels of welfare (Anang and Yeboah 2019). Thus, an understanding of the impact of socio-economic and demographic factors provides an opportunity for profiling households to determine their needs. The level of education in most rural areas of developing countries is lower than what prevails in urban areas, which makes people in rural areas less likely to be employed in high paying jobs (Anang and Yeboah 2019). However, there are several dimensions of assessing welfare. Welfare indicators generally refer to a household's knowledge of their resources including income (Ndayambaje et al. 2012), which influences specific sociocultural determinants (Delpeuch et al. 1999). When assessing the economic level of the household, it is important to observe that the variables used in computing the economic index reflect more the permanent living conditions of the households than current cash availability; a low economic level thus indicates medium to long-term poverty. Women in many rural communities engage in petty trading and other income earning activities to supplement household income (Anang and Yeboah 2019). In the context of the developing countries, where 70% of the population living in the rural areas are considered to be unemployed, their contribution to household activities is hindered (Sekhampu 2012).

## 3. Materials and Methods

### 3.1. Study Area

As detailed by Ndhlovu et al. (Ndhlovu et al. 2019), the study was conducted across 16 villages covering four municipalities in the Vhembe District Municipality, Limpopo Province of South Africa (Table 1). The study area is exceptionally endowed with numerous environmental biomes which can be used for herbal cosmetic production. Vhembe district has a landmass of 25,597 km$^2$ (Table 2), with

the majority living in villages (Statistics South Africa 2012). Vhembe District Municipality contains noteworthy biodiversity and rich heritage (Ross 2017; Nhemachena et al. 2015). All (100%) the participants in the study area belong to the Vhavenda cultural group.

**Table 1.** Selected villages in the four local municipalities of Vhembe District Municipality, Limpopo province, South Africa.

| Local Municipality | Villages |
|---|---|
| 1. Collins Chabane municipality | 1. Khakhanwa<br>2. Tondoni<br>3. Dididi<br>4.Tshikonelo |
| 2. Makhado municipality | 5. Tshakuma<br>6. Ludanani<br>7. Muhovheya<br>8. Dovhuni<br>9. Muguvhumi<br>10. Diambele |
| 3. Thulamela municipality | 11. Mphego<br>12. Tshimutikili<br>13. Levumbhi<br>14. Mukomaasaanandou<br>15 Mukula |
| 4.Musina | 16. Folovhudwe |

**Table 2.** Description of Vhembe District Municipality, Limpopo province, South Africa.

| Description | Units |
|---|---|
| Area total | 25,597 km$^2$ |
| Population total<br>Density | 1,294,722 million people<br>51/km$^2$ (130/sq. mi) |
| Racial Makeup | Black African 98.2%<br>Colored 0.1%<br>Indian/Asian 0.4%<br>White 1.1% |
| Languages | First language—Venda 67.2%<br>Tsonga 24.8%<br>Northern Sotho (Sepedi) 1.6 %<br>Other languages 5.1 % |
| Sex | Male 590,509 (45.6%)<br>Female 704,559 (54.4%) |

*3.2. Sampling Technique*

In order to have meaningful information, the data were sampled within two major sampling methods, viz. the probability (one village from each municipality was selected randomly from the four municipalities) and non-probability methods as well as purposive (expert) sampling which is a participant selection tool widely used in ethnobotany (Tongco 2007). This sampling technique is also called judgment sampling, which is a cautious choice of participants due to the qualities of the information they possess. People with a specific profile were selected in order to obtain high quality and reliable information.

For the current study, the age of the participants ranged from 20 to 80 years old, and volunteers living in a rural environment and from a variety of socio-economic strata who had knowledge of herbal plants were contacted. A total of 79 Vhavenda women who had knowledge about

herbal-based cosmetics and cosmeceuticals participated in the study. The data was collected using semi-structured questionnaire to probe questions relating to the contribution of herbal-based cosmetics and cosmeceuticals to the welfare of the Vhavenda women. Net income and profit index of knowledge holders that deal with herbal-based cosmetic and cosmeceutical were recorded.

### 3.3. Data Analysis

Descriptive statistics such as frequency counts, proportions, standard deviation and inferential statistics i.e., Tobit regression and Budgeting analysis were used to analyze and describe the socio-economic characteristics of the participants in the study area. Data analysis was conducted using SPSS version 25 and Stata SE version 11.

### 3.4. Theoretical Model and Empirical Specification

#### 3.4.1. Tobit Regression

In this study, factors influencing the income generated from herbal-based cosmetics and cosmeceuticals were analysed using Tobit regression. Of the quantitative response models on welfare economics, the Tobit regression model is a hybrid of the discrete and continuous models, one of the analytical tools that favoured this study because of its dual purpose of measuring the elasticity of the probability on the herbal-based cosmetics and cosmeceuticals household's income. Following the earlier studies (Ojimba 2013; Adelekan and Omotayo 2017; Awotide et al. 2019), the total income generated from the sales of herbal-based cosmetics and cosmeceuticals was the dependent variable. Given that the objectives of this study are beyond the determinants of a household's income, to analyze the intensity of the households' income, we therefore adopted the Tobit regression model. In addition, Tobit regression is an extension of the probit model which is useful for continuous values that are censored at or below zero as we have in this data set. When a variable is censored, regression models for truncated data provide inconsistent estimates of the parameters. The Tobit model assesses the probability of the household income, as well as the intensity or degree impact of the income obtained by women selling herbal-based cosmetics and cosmeceuticals in relation to their socio-economic and demographic characteristics. The Tobit model supposes that there is a latent unobserved variable $g_i^*$ that depends linearly on $z_i$ through a parameter vector $\alpha$. There, $\tau_i$ is a normally distributed error term to capture the random influence on this relationship. The observed variable $g_i$ is defined as being equal to the latent variable whenever the latent variable is above zero and equal to zero otherwise (1).

$$g_i = \{o_{if}^{g_i^*} \; \frac{if}{g_i^*} \; \frac{o}{\leq \; o} \tag{1}$$

where $g_i^*$ is a latent variable:

$$g_i^* = a z_i + \tau_i,$$
$$\tau_i \mathcal{N}\left(o, o^2\right)$$

If the relationship parameter $\alpha$ is estimated by regressing the observed $g_i$ on $z_i$, the resulting Ordinary Least Squares estimator (OLS) is inconsistent. Freeman et al. (Freeman et al. 1998) have proven that the likelihood estimator suggested by Tobin (Tobin 1958) for this model is consistent. The likelihood function of the model (1) is given by $L$ (Equation (2)).

$$L = \prod_O F_i\left(g_{oi}\right) \prod_i f_i(g_i) \tag{2}$$

$$L = \prod_O [1 - F(z_i a / o)] \prod_i o - f[(g_i - z_i a)/o]$$

where $f$ and $F$ are the standard normal density and cumulative distribution functions, respectively.

Then we can write the log-likelihood function (Equation (3)).

$$\log L = \sum\nolimits_0 \log(1 - F\left(\mathcal{Z}_i a / \sigma\right) + \sum\nolimits_1 \log\left(\frac{1}{(2\prod \sigma 2)^{1/2}}\right) - \sum\nolimits_1 \frac{1}{2\sigma^2}(\mathcal{G}_{i-az_i})^2 \tag{3}$$

which is there estimated by maximizing the log-likelihood function (Equation (4))

$$\begin{cases} \frac{\partial \log L}{\partial \sigma} = -\sum\nolimits_O \frac{z_i f\left(\frac{z_i \alpha}{\sigma}\right)}{1 - F\left(Z_i \frac{\alpha}{\sigma}\right)} + \frac{1}{\sigma^2}\sum\nolimits_1 (\mathcal{G}_i - \alpha z_i)z_i = 0 \\ \frac{\partial \log L}{\partial \sigma^2} = \frac{1}{2\sigma^2}\sum\nolimits_O \frac{\alpha z_i f\left(z_i \alpha / \sigma\right)}{1 - F\left(Z_i \alpha / \sigma\right)} - \frac{n_i}{2\sigma^2} + \frac{1}{2\sigma^4}\sum\nolimits_1 (\mathcal{G}_i - \alpha z_i)^2 = 0 \end{cases} \tag{4}$$

An iterative process is usually employed to obtain the maximum likelihood estimator of (Equation (4)), because they are non-linear. The variables used in the analysis are presented below. The dependent variable indicating the income (dollars) generated from herbal-based cosmetics and cosmeceuticals for the productive season under investigation. While the independent variables were experience (years), marital status, educational attainment (years), household size (actual number), age of the participants (years), teenagers, municipalities, tools used for production, market benefits, market trends, consumption patterns. Others are religious affiliation, expenditure, employment status, household size, children living in the household, produced products, teenagers, benefits of herbal-based cosmetics and cosmeceuticals, payment by consumers, participants' production cost, and consumption patterns of herbal-based cosmetics and cosmeceuticals. It was therefore hypothesized that herbal-based cosmetic and cosmeceutical income of the participants has nothing to do with the socio-economic characteristics of the study area.

### 3.4.2. Gross Margin Analysis

This was used to estimate the cost and return on herbal-based cosmetics and cosmeceuticals in the study area. The formula used for the calculations was as detailed by Daud et al. (2018). We estimate the cost and return on herbal-based cosmetic and cosmeceutical in the study area as:

$$GM = TR - TVC$$

$$\text{Benefit cos t ratio (BCR)} = \frac{TR}{TC}$$

$$TC = TFC + TVC$$

where: TC = total cost, TR = total revenue, TC = total cost, TR = total revenue, TFC = total fixed cost, TVC = total variable cost, GM = gross margin.

## 4. Results and Discussion

### 4.1. Demographic Characteristics of the Participants

In the present study, the socio-economic characteristics that were considered included age, household size, employment status, educational level, marital status and years of experience (Delpeuch et al. 1999; Celik and Hotchkiss 2000). As indicated in Table 3, the participants were grouped into four age groups and the majority (39%) of them were within the 56–70 years age group. The large number of old women in the business could translate into inability to accept or learn new methods/innovation. This implies that the old women might be unable to adopt new ideas and innovations to enhance productivity. Therefore, the legacy of the indigenous knowledge of herbal-based cosmetics and cosmeceuticals in Vhembe District Municipality is possibly in danger of being eroded with the passing away of these old women.

**Table 3.** Demography of the Vhavenda women who were knowledgeable about herbal-based cosmetics and cosmeceuticals in Vhembe District Municipality, Limpopo province, South Africa (n = 79).

| Parameter | Frequency | Percentage (%) |
|---|---|---|
| Age distribution | | |
| 26–40 years | 9 | 11 |
| 41–55 years | 25 | 32 |
| 56–70 years | 31 | 39 |
| 71 and above | 14 | 18 |
| Household size (individual/s) | | |
| 0–1 | 3 | 3.8 |
| 2–3 | 17 | 21.5 |
| 4–5 | 41 | 51.9 |
| 6 and more | 18 | 22.8 |
| Employment status | | |
| Formal employment | 4 | 5 |
| Not employed | 35 | 44 |
| Part time | 9 | 12 |
| Self employed | 11 | 14 |
| Retired | 16 | 20 |
| Volunteering | 4 | 5 |
| Education status | | |
| Informal education | 18 | 23 |
| Primary education | 24 | 30 |
| Secondary education | 27 | 34 |
| Tertiary education | 10 | 13 |

On the other hand, Arthur (2005) indicated that older people tend to adhere strictly to traditional methods of production while younger people are often willing to adopt new methods and innovation in order to increase production. A similar finding was also observed by Thondhlana and Muchapondwa (2014). Households with older members (60 years and above) were small, in part, due to children moving away to seek new opportunities in towns and cities, or to start their own households. This could also lead to a lack of local production of natural-based cosmetics and cosmeceuticals. The current finding on age distribution indicates that the majority of the participants do not fall within the category of the preferable productive age group.

In terms of the size of the household, the majority (52%) consisted of 4–5 individuals (Table 3). These results are in line with the assertion of Kyei (2011). More so, Arthur (Arthur 2005) asserted that the size of the family is of high significance for the nation in general as well as to the welfare and wellbeing of the participants, the families and community. In this study, 52% of the participants were committed to small family size (Table 3). With 22.8% of participants with a family size of over 6 people, it can be inferred that only a few of the population was in support of large families, although the fact is that 52% responded in favour of a family size of above four but fewer children. The analysis invariably showed that the participants were committed to a smaller family size.

In the current study, formal education attainment was low among the participants. For instance, 23% did not receive a formal education, while 30% attended primary school only (Table 3). In addition, 34% of the participants had secondary education while 13% attended tertiary education. According to Cameron and Harrison (2012), formal education refers to an educational model to deliver a predefined curriculum and offered by institutions in a classroom-based setting. Vhembe is one of the districts with the lowest educational attainment among individuals aged 25–64 years (Vhembe District Municipality 2016). This implies that most of the participants were informally educated; hence the chances of adopting modern technology or different methods from other cultures could be lower (Nmadu et al. 2015). This could have affected their chances of using improved

technologies which require training and the reading of manuals in order to master modern techniques of cosmetic formulation. Low levels of education negatively affect the success of small- and medium-scale enterprises and programs. Education, particularly training, enhances the adoption of technology and improved methods which are a vital means to achieving higher productivity (Organisation of Economic and Co-Operation Development (OECD) 2018). However, the current findings were different from the study by Delpeuch, Traissac, Martin-Pre'vel, Massamba and Maire (Delpeuch et al. 1999).

Furthermore, the employment status variable indicated that 44% of the participants were unemployed (Table 3). The finding of this study aligns with the government statistics on unemployment for the study area (Statistics South Africa 2018). However, most of the participants engaged in multiple occupations thereby diversifying their source of income (Kyei 2011).

### 4.2. Tobit Regression Model of Factors Affecting Income Generated from Herbal-Based Cosmetics and Cosmeceuticals by the Participants

The determinants of income of the participants was estimated using Tobit regression. The F-test result shows that the estimates of an equation of the model are jointly significant at 1%, 5% and 10% levels. In addition, the computed Likelihood Ratio Chi Square statistics are statistically significant ($p < 0.01$) in the estimated model. This implies that the estimated parameters are not jointly equal to zero. The null hypothesis set at the methodological section is hereby rejected. From the 20 variables fitted in the model, year(s) of experience, expenditure and consumption patterns were statistically significant at 1% level. Additionally, variables such as teenagers in the household, teenagers, and payment by consumers were statistically significant at 5% level; while total number of children by participants and the age of the participants were statistically significant at 10% level (Table 4).

**Table 4.** Tobit Regression Model of income determinants of the herbal-based cosmetics and cosmeceuticals by Vhavenda women in Vhembe District Municipality, Limpopo province, South Africa (n = 79).

| Parameter | Coefficient | Std. Error | t-Value | p-Value | Interval |
|---|---|---|---|---|---|
| Experience | −0.6160918 | 0.1160554 | −5.31 | 0.000 *** | −0.3829877 |
| Marital status | 0.1674765 | 0.1204115 | 1.39 | 0.170 | 0.4093302 |
| Religious affiliation | −0.6951942 | 0.3244218 | −2.14 | 0.037 ** | −0.0435738 |
| Education attainment | 0.1532906 | 0.1133938 | 1.35 | 0.183 | 0.3810487 |
| Expenditure | 0.6121074 | 0.115956 | 5.28 | 0.000 *** | 0.8450119 |
| Employment status | 0.001468 | 0.0789941 | 0.02 | 0.985 | 0.1601323 |
| Household size | 0.0261993 | 0.0857474 | 0.31 | 0.761 | 0.1984281 |
| Teenagers in the household | −0.4374788 | 0.4743294 | −0.92 | 0.361 | 0.51524 |
| Municipalities | −0.2031099 | 0.2212987 | −0.92 | 0.363 | 0.2413816 |
| Tools | −0.1488979 | 0.1018302 | −1.46 | 0.150 | 0.055634 |
| Products | −0.0635793 | 0.1074252 | −0.59 | 0.557 | 0.1521906 |
| Teenagers | 1.05755 | 0.5049819 | 2.09 | 0.041 ** | 2.071836 |
| Benefits of herbal-based cosmetic | 0.0047007 | 0.2131586 | 0.02 | 0.982 | 0.4328424 |
| Market trends | 0.0989917 | 0.0781714 | 1.27 | 0.211 | 0.2560036 |
| Payment by consumers | −0.0064162 | 0.003018 | −2.13 | 0.038 ** | −0.0003543 |
| Consumers | −0.0255346 | 0.0726553 | −0.35 | 0.727 | 0.1203979 |
| Production cost | 0.0257713 | 0.0725378 | 0.36 | 0.724 | 0.1714678 |
| Total number of children by participants | 0.3940138 | 0.226709 | 1.74 | 0.088 * | 0.8493723 |
| Age of the participants | 0.2337287 | 0.1220305 | 1.92 | 0.061 * | 0.4788343 |
| Consumption patterns | 3.156138 | 0.9300637 | 3.39 | 0.001 *** | 5.024226 |
| /sigma | 0.7798788 | 0.0779147 | | | 0.936375 |
| Number of obs | 79 | | | | |
| LR chi$^2$ | 56.55 | | | | |
| Prob > chi$^2$ | 0.0000 | | | | |
| Pseudo R$^2$ | 0.272 | | | | |
| Log likelihood | 75.592938 | | | | |

Note: ***, ** and * indicates 1%, 5% and 10% levels of significance, respectively.

There is a negative coefficient (−0.6160918), and 1% significant level of relationship between experience and the income level among the participants. This simply articulates that the higher the

years of experience the lower the income and vice versa. This is contrary to the a priori expectation of the study. Ordinarily, it is expected that the higher years of experience should lead to more income, because the participants should have acquired more experience in production and marketing which should have translated into increased income. However, this outcome might be peculiar to the study area; maybe the experiences acquired by the participants were not on the herbal cosmeceuticals, thereby negatively affecting their income from the herbal-based cosmetics and cosmeceuticals in the study area.

The religious affiliation variable of the participants has a negative coefficient (−0.6951942) and was significant at the 5% level of significance with the income level of the participants. This means that religious affiliation has a negative and significant relationship or contribution to the income from herbal-based cosmetics and cosmeceuticals. The results of the study are aligned with the previous studies (Bettendorf and Dijkgraaf 2011; Öhlmann and Hüttel 2018). These aforementioned authors opined that religious affiliation had a negative effect in the low-income and medium income countries such as South Africa.

The estimate for the expenditure of the participants was statistically significant at 1% level with a positive coefficient (0.6121074), indicating that the expenditure level of the participants stimulated an increase in their income level. As highlighted by Janvry and Sadoulet (De Janvry and Sadoulet 2001), education, gender and year of experience contribute a vital role in determining the income of the participants. This stands to corroborate the basic economic rule of income and expenditure that the more the income level of an individual the more the expenditure of such individual.

In addition, social grants have become an increasingly popular means of improving the welfare of poor households in South Africa (Satumba et al. 2017; Biyase). Social Assistance Act of 2004 provides the legal framework for the administration of social grants. One of the frameworks is that teenagers are targeted as categories of people who are vulnerable to poverty and in need of state support. In households where there are teenagers, it is applicable that they receive social grants. The coefficient of the participants' accessibility to social grants shows that it was statistically positive (0.041) and significant at a 5% level. This means that the participants whose children benefit from the monthly social grant have more income than their contemporaries whose children do not benefit from the grant. This is in line with the expected outcome of this research, as households with more social grants are expected to have more income (Nedombeloni and Oyekale 2015).

The study further showed that payment by consumers had a negative (−0.0064162) relationship and was significant at the 5% level of statistical significance with the participants' income level. This means that the payment by consumers did not have a significant relationship with income. It suggest that the participants relied on social grants as a source of income (Nedombeloni and Oyekale 2015). In the study area, "*Vhavenda aba badale"*, meaning that 'Vhavenda people do not pay' is a common phrase that is well-known among the participants. The assumption is that herbal-based cosmetics and cosmeceuticals are undermined because of lack of packaging and branding, so the consumers are not willing to pay. According to Zekiri and Hasani (2015), good packaging helps to identify and differentiate products to the consumers.

The coefficient of the total number of children by the participants was found to be positive (1.05755) and significant at a 10% level of statistical significance. This means that the number of children per household remained a key factor for bringing more income for the participants. It stands to support what was earlier discussed (concerning teenagers), that there is a positive relationship between the overall figures of children in each household income. The age parameter of the participants was also found to be positive (0.2337287) and significant at a 10% level of significance. This means that the age of the participants is related to their income level. This could probably be due to the fact that when people age they acquire more knowledge about herbal cosmetic production, hence more income. This factor was also highlighted by Adelekan and Omotayo (2017). The authors indicated that age was statistically significant to the productivity and income of farmers.

The Tobit regression fitted coefficient for consumption patterns of herbal-based cosmetics and cosmeceuticals was statistically positive (0.0257713) and significant at a 1% level of significance on the income of the participants. This indicates that the consumption rate of the participants increases with an increase in their income level. This actually validates the a priori knowledge that the consumption of individuals increases with income level (Burger et al. 2015). As highlighted by Brück (2001), income and consumption assess several dimensions of welfare; this observation aligned with the results on the current study. On the other hand, education and consumption are determinants of welfare (Celik and Hotchkiss 2000).

*4.3. Budgeting Analysis of the Response by the Participants*

Gross margin was used to estimate the cost and return on herbal-based cosmetic and cosmeceutical production in the study area as presented thus: GM = TR − TVC

$$\text{Benefit cos t ratio (BCR)} = \frac{\text{TR}}{\text{TC}}$$

$$\text{TC} = \text{TFC} + \text{TVC}$$

where: TC = total cost, TR = total revenue, TC = total cost, TR = total revenue, TFC = total fixed cost, TVC = total variable cost, GM = gross margin:

$$\text{TR} = 107.37 \text{ USD}$$

$$\text{TVC} = 53.94 \text{ USD}$$

$$\text{TFC} = 30.54 \text{ USD}$$

$$\text{TC} = 83.89 \text{ USD}$$

$$\text{GM} = 107.37 \text{ USD} - 83.89 \text{ USD} = 23.64 \text{ USD}$$

$$\text{BCR} = \frac{\text{R1841.01} \left(107.37 \text{ USD}\right)}{\text{R1438.42} \left(83.89 \text{ USD}\right)} = 1.28 \ (0.075 \text{ USD})$$

Note: USD 1 = 17.03 South African rand.

Gross margin analysis was utilized to gauge the expense and profit of herbal-based cosmetic and cosmeceutical production in the study area. Thus, the benefit cost ratio (BCR) was 1.28 (0.073 USD). By implication, given that the BCR is greater than 1, it suggests that herbal-based cosmetic and cosmeceutical production is lucrative in Vhembe District Municipality of Limpopo Province, South Africa. This implies that for every R1.00 (0.057 USD) invested in herbal-based cosmetic and cosmeceutical production in the study area, an expected return of R1.28 (0.073 USD) return will be realised ceteris paribus. The profitability of herbal-based cosmetic and cosmeceutical enterprise is expected to increase significantly if more capital is ploughed into its production. As indicated by Ladzani and Netswera (2009), skills development enhances the labour force participation in an economy, as it assists in reducing unemployment, inequality and poverty that continues to be essential challenges in rural areas. Although some support institutions/agencies are currently in place, the awareness is relatively low among emerging rural entrepreneurs. Thus, government and other support agencies need to intensify efforts to raise the awareness and promotion, especially for herbal-based cosmetic and cosmeceutical production in local areas (Rampedi 2010).

## 5. Conclusions

The current findings established the link between the environmentally endowed herbal-based cosmetics and cosmeceuticals and the welfare of the Vhavenda women in Vhembe District Municipality. The majority of the participants were over 60 years old, which is an indication that the

knowledge is held by the older generation. Educational attainment was found to be the key significant variable in the descriptive and inferential model statistics, which implies that formal education should be strongly encouraged as a means to improve their livelihood. Variables such as years of experience in herbal-based cosmetics and cosmeceuticals, expenditure levels of the participants, teenagers in the household, total number of children by participants, age of the participants, and consumption patterns were the socio-economic independent variables that were statistically significant to income level from herbal-based cosmetics and cosmeceuticals. A budgeting cost ratio indicates that for every R1.00 (0.057 USD) invested in the herbal-based cosmetic and cosmeceutical production, an expected return of R1.28 (0.073 USD) was realised. However, the study excluded the discount rate and time range in budgeting analysis (benefit cost ratio) due to the limited duration (1 year) of the research. Taken together, these findings serve as inputs for the evidence-based policy interventions to promote herbal-based cosmetics and cosmeceuticals as well as radical socio-economic transformation, particularly in the study area. The policy makers should encourage the youth in rural communities by implementing policies and providing incentives that will make herbal-based cosmetics and cosmeceuticals more lucrative. In addition, an in-depth inventory of herbal-based cosmetics and cosmeceuticals with associated economic potential should be taken with a more deliberate effort by the government, researchers and non-governmental organizations. This may be facilitated by increased funding and active promotion of herbal-based cosmetics and cosmeceuticals enterprise in the region.

**Author Contributions:** Conceptualization, P.T.N., A.O.O., A.O.A. and W.O.-M.; Formal analysis, P.T.N. and O.A.; Funding acquisition, W.O.-M.; Investigation, P.T.N.; Project administration, W.O.-M.; Resources, W.O.-M.; Supervision, W.O.-M.; Validation, A.O.A.; Writing—original draft, P.T.N.; Writing review and editing, A.O.O., A.O.A. and W.O.-M. All authors have read and agreed to the published this version of the manuscript.

**Funding:** This research was funded by the National Research Foundation, Pretoria, South Africa (grant number: UID 105161). The Article Processing Charge (APC) was funded by the Faculty of Natural and Agricultural Sciences, North-West University, South Africa.

**Acknowledgments:** We are grateful to all the knowledge holders in selected villages of Vhembe District Municipality who willingly participated in the project. We thank Nkhanyseni Thukutha and Luesani Tshikovha (research assistants) for their great contribution during the field work.

**Conflicts of Interest:** The authors declare no conflict of interest.

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
