# Peer review of "Herbal-Based Cosmeceuticals and Economic Sustainability among Women in South African Rural Communities"

_economies, doi:10.3390/economies8030051_

Round 1

Reviewer 1 Report

The article is quite interesting. The introduction should be expanded to include the latest items in world literature. A substantive discussion of the results should be carried out.

Reviewer 2 Report

The article addresses an interesting and valuable topic. Welfare status analysis among the rural population is particularly significant due to the problems of economic and social sustainability in agricultural areas. Therefore, the article arouses interest and encourages discussion. The authors correctly presented the concept of the study and described the research methods. I have no objections as to how the data were collected and the descriptive statistics presented. This part of the publication is fully acceptable.
Critical remarks refer to the part related to measuring the impact of selected factors on welfare status. First of all, why expenditure was chosen to measure welfare status. A better and more frequently used variable in this case is household income. Expenditure is a derivative of income, although it does not have to be an actual reflection of the income situation. After all, there are different rates of marginal propensity to consume / save between individual units. In addition, if Friedman's theory of permanent income was taken into account, current income does not directly translate into expenditure over the same period. And yet the article is about indicating the determinants of the life situation of the surveyed women here and now.
It is also unknown why the authors in one analysis - OLS - take expenditure as a dependent variable and income is an explanatory variable, and in the other - Tobit regression - the relationship is described in a completely different way. As a result, we don't know whether income affects expenditure (OLS) or expenditure has an influance on income (Tobit). Anyway, in the latter case, the relationship is apparent, because expenditure is a derivative of income, not the other way around.
Taking this approach raises many problems with inference. For example, in the OLS model the variable "experience" has a positive effect on welfare state, while in the Tobit model it has a negative sign. And yet welfare state improves through higher incomes. Moreover, the authors indicate that the relationship between experience and income is positive: "As highlighted by Janvry and Sadoulet, education, gender and year of experience contribute a vital role in determining income of the participants." (line 288-289) How do you explain this contradiction? Similarly conflicting results were obtained for the variable "religious affiliation". In the Tobit model it had a negative sign, while earlier in OLS it had a positive effect on welfare state. Finally, the Tobit model indicates that "the payments by consumers" variable has a negative effect on income. How does this apply to budgeting analysis, the results of which suggest that the business is profitable. If it were stated that "the payment by consumers did not have a significant relationship with income" (line 304) and "the consumers are not willing to pay" (line 308), would profitability be guaranteed? Such inference introduces a lot of chaos and it is not finally known what dependencies occur in practice.
Hence, I suggest that you decide on one model in which the dependent variable will be income. The more so that the other explanatory variables can be treated as income determinants. Anyway, in one part - gross margin analysis - an income approach was used for the analysis, so it would be consistent across the entire article.
With additional comments:
- I propose to supplement the authors' assumptions regarding the expected analysis results in the "Materials and methods" section. Then the results will confirm the hypotheses or reject them
- lines 209-211 - Authors state the obvious, there is nothing insightful in saying that higher income improves welfare state. Attention should rather be focused on explaining what factors shape this income, and therefore welfare state
- the authors mistakenly use the terms "revenue" and "income" to describe the same variable. In this case, it's probably about "income"? (see lines 132-133 or 264-266)
- I propose that the values ​​of income, expenses, etc. should be presented in dollar, euro (or other world currency), because for readers outside of South Africa the national rate is not known,
- adjust the keywords, they currently do not fully reflect the content of the publication
- please explain what the authors understand by the term "consumption patters" as a variable in the analysis
- how is the employment status variable interpreted? Should employment improve welfare status or unemployment?
- please describe the discount rate and time range in budgeting analysis (bebefit cost ratio)

Reviewer 3 Report

The paper is very interesting as it explores the conditions on which environmental assets could be used as a means to enhance the well-being of indigenous communities. Nevertheless, the theoretical foundation of the paper is so weak that leads me to reject it at its current form. Please see some comments that may help you improve your paper.

1) The paper lacks any global research question. The understanding of the factors influencing the well-being of Vhavenda women is of little interest for the scientific community unless the results were targeted to answer a more general question (i.e. how women in indigenous communities earn their income or capitalize their creativity and skills. The role of youth in supporting local economies. Rurality vs sustainability etc).

2) The absence of any general theoretical discussion becomes more evident when authors discuss their methods and variables. There is no reasoning and theoretical evidence for using these variables and no further discussion of how the findings could enhance our knowledge around a possible research question. By this way, all results are heavily context dependent and of limited interest for the wider scientific community. Moreover, there is no discussion about the selection of the methods. For example, why using OLS and Tobit and not only OLS.

Since authors have put a significant effort on completing their survey and they showed a good capacity in conducting quantitative research I would encourage them to resubmit their manuscript. I choose rejection and not major revisions because I want to give them time to elaborate on their research questions. In case they want to resubmit their paper, I would be more than glad to review it again, under the condition that they will elaborate on the following issues.

Please check your questions in the survey and try to formulate some research questions. Try to imagine a wider issue to confront (one that would be interesting for a reader residing thousand miles away from South Africa) and not just to reveal contextual relationships. This will become feasible when you will have reviewed the relevant literature. You will understand when your research question is interesting when after your empirical analysis you will be in position to discuss how your results question or validate previous findings and add to the existing knowledge. Good luck.

Round 2

Reviewer 2 Report

Thank you for responding to my comments. The corrections made by the authors and the explanations contained in the letter are convincing. In its current form, the article meets the requirements of scientific work and can be published.

Reviewer 3 Report

The paper has been improved. Therefore, I support its publication in the journal. Good luck with this and any future papers.

This manuscript is a resubmission of an earlier submission. The following is a list of the peer review reports and author responses from that submission.